# Experience with the mTOR Inhibitor Everolimus in Pediatric Liver Graft Recipients

**DOI:** 10.3390/children10020367

**Published:** 2023-02-13

**Authors:** Mathis Wehming, Dorothée Krebs-Schmitt, Andrea Briem-Richter, Bianca Hegen, Florian Brinkert, Lutz Fischer, Enke Grabhorn

**Affiliations:** 1Department of Pediatric Gastroenterology and Hepatology, University Hospital Hamburg-Eppendorf, 20246 Hamburg, Germany; 2Department of Hepatobiliary Surgery and Transplantation, University Hospital Hamburg-Eppendorf, 20246 Hamburg, Germany

**Keywords:** pediatric liver transplantation, graft dysfunction, immunosuppressive therapy, renal impairment, malignancy

## Abstract

Introduction: Immunosuppression after pediatric liver transplantation remains a major challenge. MTOR inhibitors provide a promising therapeutic approach in combination with reduced CNI after transplantation. However, there are still few data regarding their use in children. Patients: We analyzed 37 patients with a median age of 10 years, who received Everolimus for one or more of the following indications: I = chronic graft dysfunction (*n* = 22); II = progressive renal impairment (*n* = 5); III = non-tolerable side effects with previous immunosuppressive medication (*n* = 6); and IV = malignancies (*n* = 10). The median follow-up time was 36 months. Results: Patient survival was 97%, and graft survival 84%, respectively. Stabilization of graft function was observed in 59% in subgroup 1, with 18.2% ultimately requiring retransplantation. No patient in subgroup IV developed recurrence of his primary tumor or PTLD by the endpoint of the study. Side effects were observed in 67.5% of the study patients, with infections being the most frequent (*n* = 20; 54.1%). There were no relevant effects on growth and development. Conclusion: Everolimus seems to be a treatment option in selected pediatric liver graft recipients for whom other regimens are not suitable. Overall, the efficacy was good and the side effect profile appeared to be acceptable.

## 1. Introduction

The long-term outcome after pediatric liver transplantation (LT) has improved dramatically in the last few decades [1,2]. Although there seems to be a number of patients who can wean off immunosuppressive drugs in the long-term follow-up [3], most patients depend on life-long medication to prevent graft rejection. Standard immunosuppressive regimens after LT are usually based on calcineurin inhibitors (CNIs) with typical side effects comprising kidney dysfunction [4] and hepatotoxicity [5,6]. Furthermore, occurrence of post-transplant lymphoproliferative disease (PTLD) can compromise the outcome [7], especially in children with risk factors such as young age (<10 years) and EBV negativity prior to LT [8]. About 60% of late deaths after pediatric LT can be attributed to immunosuppressive complications [9]. Moreover, some patients develop chronic graft rejection with severe fibrosis and graft dysfunction despite dual or triple immunosuppressive therapy [10]. CNI-related nephrotoxicity is responsible for renal dysfunction in up to 30% of pediatric patients more than 10 years after LT [11,12] and has been associated with significant morbidity [13,14]. Since pediatric patients have longer exposure to immunosuppressive therapy and may move from mild asymptomatic impairment to symptomatic end-stage renal failure that requires kidney transplantation, it is essential to be aware of this deterioration and switch to CNI-sparing treatment options. The mTOR inhibitor (mammalian target-of-rapamycin inhibitor) Everolimus (EVL) is an antiproliferative immunosuppressive drug without significant nephrotoxicity, but with side effects such as mouth ulcerations, dyslipidemia and impaired wound healing [15]. Data on adult transplantation show that EVL in combination with CNIs can maintain graft function in de novo and maintenance liver transplant recipients [16,17] while improving or preserving renal function. Due to the synergism of CNIs and mTOR inhibitors, the dosage of CNIs can be significantly reduced [18]. Data in pediatric LT recipients are rare. Gibelli et al. reported that conversion from Tacrolimus (Tac) to Sirolimus is safe in selected pediatric patients [19]. Nielsen et al. found promising results with EVL in 18 pediatric patients with chronic graft dysfunction, CNI toxicity and malignant disease [20]. The results of a multicenter, single-arm, prospective study recently demonstrated improved renal function with early introduction of EVL and reduced CNI after pediatric LT, but the results also raised safety concerns regarding over-immunosuppression [21].

We present our data on 37 pediatric patients who received EVL either de novo or as maintenance therapy after LT. We aimed to evaluate the efficacy, safety and tolerability of this therapy.

## 2. Materials and Methods

### 2.1. Patients

From 1992 to 2014, 744 pediatric LTs were performed at our transplant center. We performed a retrospective chart analysis and identified 37 pediatric patients who were treated off-label with EVL after informed consent of the parents. Indications were either failure of the preceding immunosuppressive treatment with chronic rejection (subgroup I), potentially previous CNI treatment-related side effects such as impaired kidney function (subgroup II) or others (subgroup III), and the antiproliferative effect of mTOR inhibitors in case of underlying malignant diseases (subgroup IV). Five patients received EVL for several concomitant indications and the therapeutic outcomes were evaluated separately in each group. The date on which therapy with EVL started was determined as the baseline.

### 2.2. Standard Immunosuppressive Medication

Standard immunosuppression after LT at our center was CNI-based and consisted of the CNI Cyclosporine A (CsA; initial trough levels 150–170 µg/L, maintenance levels after 1 year 100 µg/L) or Tac (initial trough levels 7–9 µg/L; maintenance levels after one year 5–6 µg/L) and corticosteroids with an initial dose of 15 mg/m^2^. Steroids were usually stopped after one year. In all patients, the anti-interleukin-2 receptor antibody Basiliximab was administered in two single doses on day 0 and day 4 post-transplant. Tumor patients were treated de novo with EVL in combination with reduced CNI doses (*n* = 9, initial target trough levels: EVL 4–6 µg/L; CsA 80–100 µg/L; Tac 4–6 µg/L; maintenance levels: EVL 3–6 µg/L; CsA 40–60 µg/L; Tac 3–5 µg/L).

## 3. Results

### 3.1. Study Population

The mean patient age at the start of therapy was 9 years (range: 7 months to 18 years), and the median time between LT and start of EVL treatment was 23 months (range: 6 days to 194 months [16 years]). The median follow-up time was 36 months (range: 12 to 60 months). A total of 22 patients were considered maintenance graft recipients and 15 patients de novo graft recipients (time between LT and start of EVL > 6 months and ≤6 months, respectively). The main indication for LT was biliary atresia (*n* = 12; 32%), followed by metabolic diseases (*n* = 6; 16%), familial cholestatic syndromes (*n* = 4; 11%), hepatoblastoma (*n* = 4; 11%), and others (*n* = 11; 30%).

The majority of patients were switched to EVL for chronic graft dysfunction (subgroup I: *n* = 22; 59%). Other indications for modification were progressive decrease in the eGFR (subgroup II: *n* = 5), intolerable side effects other than nephrotoxicity (subgroup III: *n* = 6), and malignant diseases (subgroup IV: *n* = 9), including malignant diseases leading to LT (*n* = 7) as well as PLTD (*n* = 2). Baseline data are summarized in Table 1. The patients’ baseline characteristics for the four subgroups are listed in Table 2.

### 3.2. Immunosuppression

Patients received EVL in combination with reduced CNI (89.2%; CsA: *n* = 23, Tac: *n* = 10) or as monotherapy (10.8%; *n* = 4). Reasons for monotherapy were contraindications for CNIs (*n* = 3) and treatment of PTLD after chemotherapy (*n* = 1). In two patients, the CNI was switched from CsA to Tac after 6 weeks and 6 months post-baseline, respectively, due to occurrence of biopsy-proven acute rejection (BPAR). In one case, retransplantation was required later on due to ongoing graft failure. Another patient was switched back from Tac to CsA after 12 months post-baseline due to severe EBV reactivation. Trough levels of EVL, CsA and Tac remained in the target range throughout the study (Figure 1).

#### 3.2.1. Efficacy

Baseline biopsies were performed in 16 of 22 patients (72.7%) in subgroup I. Most patients had moderate rejections with a Banff score of more than 4/9 (*n* = 13); three patients had milder graft rejections. Biopsy post-baseline was available in 31 patients (subgroup I; *n* = 19). In 6 patients (subgroup I; *n* = 3), suitable histology data were missing because of premature withdrawal of EVL due to side effects (*n* = 3) and missing consent for follow-up biopsies. These cases were excluded from our analysis.

Overall, 32.3% (10 out of 31 patients) had an episode of BPAR during the observation period, consisting of 3/13 de novo patients (23%) and 7/18 maintenance patients (38.9%). Treatment rate of BPAR (tBPAR) was 100% and consisted of high-dose steroids. The rejection rate in subgroup 1 (36.8%; 7/19) was higher compared to patients receiving EVL for other indications (25%; 3/12). The rejection rate for the patients receiving EVL monotherapy was 25% (1/4). The majority of rejections were graded moderate with a Banff score of 5/9 (*n* = 4) or 6/9 (*n* = 3). In another three patients, rejections were milder (Banff score: 3/9; *n* = 2 and Banff score 4/9; *n* = 1). Retransplantation was necessary in five cases (subgroup I; *n* = 4) due to chronic graft dysfunction (range 8–31 months post-baseline). Of these patients, two continued treatment with EVL after re-LT. Regarding the other patients with rejection, one receiving EVL monotherapy was treated by adding MMF. Another patient was treated by changing the CNI from CsA to Tac. Raising the EVL maintenance dose with higher trough levels was successful in another two patients with rejections. Twenty-one patients (67.7%) including 12 patients (63.2%) in subgroup I showed no signs of rejection.

Subgroup III (*n* = 6) comprised patients suffering from CNI-related side effects other than nephrotoxicity. Stabilization of leucocyte serum levels could not be achieved in two of three cases, resulting in the discontinuation of EVL. The therapeutic approach in the third case was successful. Progression of gingival hyperplasia was stopped after reduction in CsA and supplementation of EVL. Conversion to an EVL-based regimen was also able to lessen symptoms in the patient suffering from polyneuropathy. Treatment of hearing impairment by switching the medication to EVL monotherapy and discontinuing CsA was not successful, leading to the need for a cochlear implant.

#### 3.2.2. Renal Function

Kidney function was monitored with the cystatin-C-based GFR formula and the Schwartz formula in combination with creatinine serum levels. The mean eGFR (SD) at baseline for patients with available data was 104.2 (±41.4) mL/min/1.73 m^2^ (*n* = 24) and 88.0 (±29.4) mL/min/1.73 m^2^ at last observation (*n* = 33). In total, nine patients completed the study with improved renal outcome. The mean eGFR increased from 86.8 (±25.8) mL/min/1.73 m^2^ to 106.2 (±33.3) mL/min/1.73 m^2^ in these patients. In the five patients of subgroup II with nephrotoxicity, the median EVL treatment duration was 36 months (range 32–60 months). Mean eGFR at baseline was 90.2 (±12.6) mL/min/1.73 m^2^, which decreased to 65.4 (±18.0) mL/min/1.73 m^2^ at the study’s endpoint. The course of eGFR is summarized in Figure 2.

#### 3.2.3. Malignancy, CMV and EBV Status

No cases of PTLD occurred in our patient collective during the observation period. No patients in subgroup IV, who received EVL for its antiproliferative effects, developed metastases or experienced recurrence of their primary tumor by the end of the study.

Twenty-six patients (70.3%) were CMV-seropositive at baseline. CMV reactivation was observed in nine patients (24.3%); two patients (5.4%) developed a mild primary infection.

Most patients were EBV-seropositive at baseline (89.2%). EBV reactivation was observed in nine patients (24.3%), while two patients (5.4%) developed a primary infection with no further reactivation. One patient suffered from severe systemic hypereosinophilic syndrome during a reactivation episode of EBV and was successfully treated with steroids. Another patient developed EBV-associated autoimmune hepatitis with acute liver failure three months after discontinuation of EVL and was treated with steroids followed by maintenance treatment with azathioprine, CsA and low-dose steroids. In the follow-up, the patient subsequently developed HHV-8-associated Kaposi sarcoma of the gastrointestinal tract and liver followed by transplant failure. He was finally retransplanted successfully and is alive with a functioning graft 4 years later on dual therapy with CsA and EVL.

#### 3.2.4. Overall Safety and Side Effects

In total, 27 out of 37 patients (73%) experienced one or more adverse events, including BPAR, graft loss or side effects. Out of 37 patients, 25 (67.5%) suffered from one or more side effects. Most side effects were infections, aphthous mouth ulceration and cytopenia. the side effects are shown in Table 3; their occurrence for each patient is shown in Table 2. Survival rate was 97.3%. One patient (2.7%) died 36 months after the start of therapy in the course of severe varicella sepsis. Overall graft survival was 86.5%.

We had to stop treatment with EVL in a total of 14 children (38%) due to side effects or failure of therapy, whilst the other 23 (62%) remained under EVL treatment.

#### 3.2.5. Growth and Development

The mean percentile at baseline (SD) was 40.1 (±31.0) for height and 37.7 (±27.9) for weight. At the study’s endpoint, the mean percentile was 37.7 (±31.6) for height and 38.5 (±29.6) for weight, showing no significant impact of EVL on these parameters.

## 4. Discussion

There are promising data for the use of EVL in adult LT recipients. However, only few data concerning the use of EVL in pediatric LT have been published so far.

In this study, we report 37 patients treated with EVL at different ages and for various reasons. We chose EVL over sirolimus due to its favorable pharmacokinetic characteristics, including faster absorption, higher bioavailability and shorter half-life [22].

Initial results with EVL in pediatric liver graft recipients were promising. Nielsen et al. treated 12 children suffering from chronic graft dysfunction with EVL in combination with CNI [20]. In total, three patients (25%) recovered normal liver function whilst partial improvement was observed in another six (50%). Recently published data from a multicenter study reported a low rate of composite efficacy endpoint incidents (treated BPAR, graft loss, death) after 12 (1.9%) and 24 (5.9%) months in pediatric LT recipients treated with EVL [21]. Adverse events occurred in all patients (100%). Comparing those findings to ours, we found a similar outcome regarding chronic graft disease. Liver graft function could be stabilized or improved in the majority of our patients, especially in those already suffering from impaired graft function. This is of note as EVL was already used as a rescue therapy in those with progressing graft dysfunction.

A major disadvantage of common immunosuppressive regimens, based on CNI, is nephrotoxicity leading to impaired renal function. Various studies have shown an improvement of renal function in adult liver graft recipients after early conversion to EVL [23], as well as use in de novo therapy [24]. Clinical trials of pediatric kidney transplantation with mTOR inhibitors in de novo treatment [25,26] along with conversion in maintenance patients [27] showed comparable findings in children. Similar results were found in studies of maintenance pediatric liver transplant recipients with either EVL or sirolimus [21,28]. These results support the assumption that mTOR inhibitors can preserve renal function, especially in patients who are at risk. We cannot confirm these results due to the diverse outcomes in our patients. The protective factor of EVL was not convincing in maintenance patients who were switched based on renal impairment. Regarding the patients of subgroup III (side effects of previous CNI treatment other than nephrotoxicity), a switch to EVL was successful in 50% of patients. Hearing loss in liver graft recipients appears to be an underestimated side effect of perioperative antibiotic and diuretic treatment together with CNI [29]. One of our patients with hearing impairment showed no improvement after conversion to EVL and reduced CsA since the impairment was already irreversible.

Although EVL may alleviate, entirely or in part, some complications occurring with other immunosuppressive regimens, it has side effects of its own [30]. Common ones are infections, hyperlipidemia, peripheral edema, mouth ulceration, angioedema, wound-healing disorders and proteinuria. Most of these can be managed with standard therapeutic measures [31]. A randomized controlled trial was performed by De Simone et al. in adult liver graft recipients who were converted from a CNI-based to an EVL-based immunosuppressive regimen 12 to 60 months after LT. Although a higher number of efficacy events, including BPAR, graft loss and death, were reported in the EVL arm (8.4%) than in the control arm (4.1%), the overall rate of those events was low [15]. However, the overall occurrence of adverse events was significantly higher in the EVL arm (95.8%) than in the control arm (69.9%).

We saw a high number of adverse events as well. Although high in number, most of them were relatively mild in our study group. Overall, the side effect profile of EVL appears to be acceptable, given its beneficial effects. Recently published data in pediatric kidney transplantation even imply a fairly low risk for developing infections associated with the use of an mTOR inhibitor in combination with low-dose CNI in de novo organ recipients [25,32]. However, we lost one patient, who was treated at another hospital for varicella sepsis. In our opinion, it is of the utmost importance to interrupt or reduce therapy with EVL in cases of severe infections, and restart after recovery.

Occurrence of PTLD is a major concern in patients receiving immunosuppression, especially in children [7]. Providing sufficient immunosuppression whilst minimizing the risks of malignancy is a balancing act due to increased de novo and recurrence rates of proliferative diseases after transplantation [33]. The reported rate of PTLD occurrence in children ranges from 6.3 to 9.7%, depending on the center’s immunosuppressive regimen and EBV status at PLT. Because of its antiproliferative properties, EVL is expected to have a protective function regarding the development of PTLD. However, Ganschow et al. reported 5 cases (8.9%) of PTLD in their study cohort, all treated de novo with EVL and reduced Tac, in a collective of 53 pediatric liver graft recipients during an observation period of 24 months [21]. Three of these cases occurred in children < 2 years old and all cases were EBV-related. Overall, 18 primary infections with EBV and 10 reactivations occurred. However, we did not observe any case of PTLD throughout the study period, despite documented EBV reactivation in nine cases (24.3%) and primary EBV infection in two cases (5.4%).

In our study, particular attention was given to patients with tumor disease that led to or occurred after LT. Nine children and adolescents were treated with EVL for its antiproliferative effect while preserving liver graft function, as described by other groups [34]. None of our nine patients developed metastases or recurrence of their primary tumor throughout the period under review. Due to the small study cohort and the fairly short duration observed, we have to point out the limitations of these results. However, considering recently published trials, it appears advisable to take mTORinhibitor-based immunosuppression into account when treating transplant recipients at risk of proliferative diseases.

As mTOR inhibitors are antiproliferative drugs, they might impair physical development. However, there was no evidence of negative effects in our study population. Data from other studies involving liver and kidney graft recipients support these findings [21,26].

## 5. Conclusions

Overall, EVL improved the outcomes of most of our patients concerning liver graft function and prevented the recurrence of tumor activity. The immunosuppression was safe. The results of our retrospective study suggest that EVL has the potential to complement established immunosuppressive regimens after PLT.

## Figures and Tables

**Figure 1 children-10-00367-f001:**
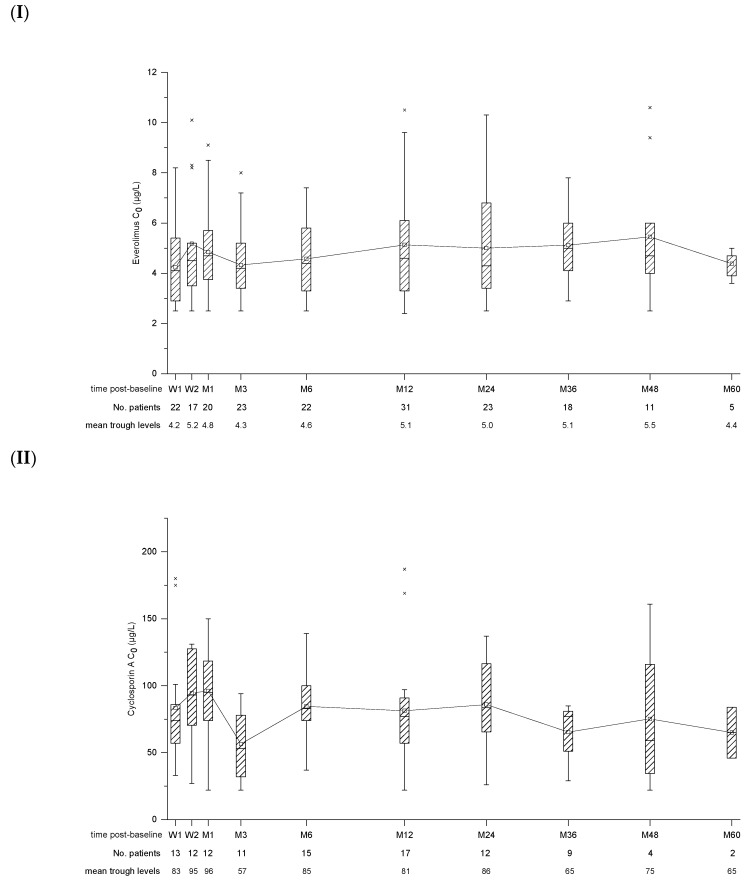
Box and whisker plots for trough concentration (C_0_) of Everolimus (**I**), Cyclosporin A (**II**) and Tacrolimus (**III**). Time post-baseline is presented in weeks (W) and months (M). The number of patients with available data as well as mean trough levels for each point of interest is displayed below.

**Figure 2 children-10-00367-f002:**
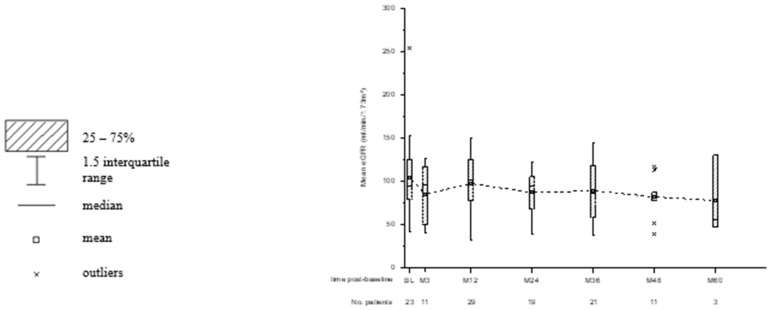
Box and whisker plots for the observed course of eGFR. Time post-baseline is presented in weeks (W) and months (M). The number of patients with available data for each point of interest is displayed below.

**Table 1 children-10-00367-t001:** Patient data.

Patient	Sex	Diagnosis Led to LT	Age at LT; Months	Age at EVL Start; Years	Previous Regime	Current Regime	Reasons for Discontinuation of Therapy with EVL	Side Effects	Follow-Up Period; Months
Group I: Chronic graft dysfunction									
1	f	biliary atresia	19	10	Tac	EVL, Tac			48
2	f	DGUOK	3	2	Tac	EVL, CsA		bi, hl	60
3	m	hepatoblastoma	162	13	CsA	EVL, Tac/EVL CsA		c	36
4	m	biliary atresia	115	12	Tac	EVL, Tac		bi	48
5	f	PSC	155	14	CsA	EVL, CsA	n/a		36
6	m	primary hyperoxaluria I	18	1	Tac	EVL, Tac			60
7	f	biliary atresia	34	12	CsA	EVL, CsA		bi, c	60
8	f	biliary atresia	104	8	Tac	EVL, Tac	progressive rejection	bi, pu	12
9	m	biliary atresia	< 1	5	CsA	EVL, CsA			12
10	f	liver cirrhosis/M. Wilson	135	13	Tac	EVL, CsA	progressive rejection, Re-LT	bi, pu	36
11	f	biliary atresia	24	5	CsA	EVL, CsA		bi, amu	15
12 *	m	CDG II	44	4	Tac	EVL, CsA	side effects		36
13	f	biliary atresia	69	12	Tac	EVL, Tac	wound-healing disorder	whd	12
14	m	n/a	160	14	Tac	EVL, CsA	progressive rejection, Re-LT	bi	12
15	m	PFIC I	44	4	Tac	EVL, CsA	side effects	bi, amu	13
16 ^I,II^	m	biliary atresia	13	10	CsA	EVL, CsA	side effects	bi, c, ae	32
17	m	PFIC II	52	4	CsA	EVL, Tac	bone marrow transplantation	bi, c, le	18
18	m	PFIC II	15	4	CsA	EVL, CsA	n/a	le	18
19	m	CPS I deficiency	28	5	CsA	EVL, CsA	side effects	bi, amu	36
20 ^I,II,III^	m	acute liver failure after enterovirus infection	19	7	CsA	EVL			60
21 ^I,III^	m	acute liver failure of unknown origin	24	12	CsA	EVL, CsA		c	12
Group II: CNI-induced nephrotoxicity									
22	m	biliary atresia	6	11	CsA	EVL			36
23	f	biliary atresia	4	16	CsA	EVL, CsA		bi, hl	48
24	m	Alagille-Syndrome	31	17	Tac	EVL, Tac			36
Group III: Side effects other than CNI-induced nephrotoxicity									
25	m	acute liver failure of unknown origin	192	16	Tac	EVL, CsA	side effects	bi, c	12
26	f	propionic acidemia	213	18	Tac	EVL, CsA/EVL, Tac			48
27	f	biliary atresia	143	13	CsA	EVL, CsA	n/a		36
28	m	connatal CMV infection	112	9	CsA	EVL		c	60
Group IV: Proliferative diseases									
29	m	hepatoblastoma	30	2	CsA	EVL, CsA/EVL, Tac	side effects	bi	12
30	m	hepatoblastoma	182	15	CsA	EVL, CsA		bi, amu	36
31	f	veno-occlusive-disease	14	1	CsA	EVL, CsA			12
32	m	hepatoblastoma	45	3	CsA	EVL, CsA		bi	48
33 ^III,IV^	m	undifferentiated hepatocellular tumor	222	18	Tac	EVL, Tac		bi, c	24
34	m	malignant epithelioid hemangioendothelioma	159	13	CsA	EVL, Tac			12
35	m	neuroblastoma with liver metastasis	3	0	CsA	EVL, CsA		bi	36
36 ^I,IV^	m	biliary atresia	7	15	Tac	EVL		bi, c, amu	48
37	m	maple syrup urine disease	2	4	CsA	EVL, CsA		bi, amu	48

^I,II,III,IV^: Displayed if part of multiple groups. bi = bacterial infection, c = cytopenia, amu = aphthous mouth ulceration, hl = hyperlipidemia, le = local edema, pu = proteinuria, ae = angioedema, whd = wound-healing disorder, DGUOK = Deoxyguanosine Kinase Deficiency, PSC = Primary Sclerosing Cholangitis, CDG = Congenital Disorder of Glycosylation, PFIC = Progressive Familial Intrahepatic Cholestasis. * = deceased during study period.

**Table 2 children-10-00367-t002:** Baseline characteristics.

Mean Age, years (SD)	9.2 (5.5)
sex	
male, n (%)	25 (68)
female, n (%)	12 (32)
height, cm (SD)	132 (33.9)
weight, kg (SD)	32.9 (18.3)
eGFR, mL/min/1.73 m² (SD)	104.2 (41.4)
CNI at baseline, n (%)	
CsA	23 (62.8)
Tac	10 (27)
none	4 (10.8)
CNI C_0_ at baseline, µg/mL (SD)	
CsA	87.9 (34.7)
Tac	6.2 (2.6)
transplantation mode, n (%)	
whole deceased liver	11 (29.7)
split organ of deceased donor (left lateral or extended right lobe)	18 (48.6)
living donor (left lateral liver)	8 (21.6)
Epstein-Barr-virus-negative, n (%)	4 (10.8)
Cytomegalovirus-positive, n (%)	26 (70.3)
induction therapy, n (%)	37 (100)
Diabetes mellitus at baseline, n (%)	0 (0)

**Table 3 children-10-00367-t003:** Side effects.

bacterial infection, n (%)	20 (54.1)
cytopenia, n (%)	9 (24.3)
aphthous mouth ulceration, n (%)	7 (18.9)
hyperlipidemia, n (%)	2 (5.4)
local edema, n (%)	2 (5.4)
proteinuria, n (%)	2 (5.4)
angioedema, n (%)	1 (2.7)
Wound-healing disorder, n (%)	1 (2.7)

## Data Availability

Not applicable.

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
