# Peer review of "Experience with the mTOR Inhibitor Everolimus in Pediatric Liver Graft Recipients"

_children, 2023, doi:10.3390/children10020367_

Round 1

Reviewer 1 Report

Very good work showing the experience of using EVL in children for LT.

I do not have any major comments about the content, but I find the figures hard to read; maybe the authors should enhance the quality. There is a little mistake with the numbering of fig 1.

Author Response

Dear Reviewer,

thank you very much for your comments. We will provide better quality figures in the final version and correct the mistake with the numbers in figure 1.

Kind regards,

Enke Grabhorn

Reviewer 2 Report

This retrospective study of experience with the everolimus in pediatric liver transplant recipients was carried out and analyzed by various indications. This study could be one of the data for adding evidence to management for Post-LT pediatric patients. The purpose of this study is well-designed, but this paper exposes significant issues in the data expression and interpretation.

More specific comments are below.

3.1. Study population

Whether nephrotoxicity after liver transplantation is secondary to the actions of CsA and Tac could be challenging to diagnose, especially among those patients with previous chronic liver disease or hepatorenal syndrome.

How are the subgroup II patients classified as calcineurin inhibitor nephrotoxicity? What is the diagnostic method or hallmark of these patients? Please clarify the diagnostic approach for CNI nephrotoxicity.

Table 1.

Discontinuation of therapy” variable need to be explained more accurately. Discontinuation of which therapy? CNI or mTORi?

Table 2.

The expression of ‘technically modified’ is a bit vague. Please re-classify as like these (for example, whole deceased liver, deceased split right liver, deceased split left liver, living right liver, the living left liver, the living left lateral liver, or variants of modified graft, etc.).

Cytomeglaievirus-positive -> Cytomegalovirus-positive

Page 7, line 14

From the manuscript, among the five patients with graft dysfunction, one patient receiving EVL monotherapy was treated by adding MMF. Another patient was treated by changing the CNI from CsA to Tac. Did modification of immunosuppressant recover these two patients' liver function without re-LT after listing up for re-LT? This paragraph is hard to understand.

Author Response

Dear reviewer,

thank you for your helpful comments to improve our manuscript. We hope, we have addressed your issues properly.

3.1. Study population with nephrotoxicity: The kidney function gradually deteriorated over time under CNI-therapy after LT. However, you are definitely right, that there exist no histological diagnosis. Therefore, we changed the description in the text. We hope, you find this more suitable.

Table 1: discontinuation of EVL was clarified in the table

Table 2: mode of transplant was specified

Page 7, line 14: text was corrected for clarification

Thank you very much again and kind regards,

Enke Grabhorn

Round 2

Reviewer 2 Report

Thank you for correcting the text in such a short period of time.

If you address inconsistent word usage and a few grammatical errors, the quality of the paper will likely improve.

The font size for each author and affiliation is inconsistent.

It is recommended to use a consistent term for mTOR throughout the paper, as it is expressed as "MTOR" only in the abstract.

Please check the numbering of (I), (II), and (III) in Figure 1.

Line 198:

2 patients (5.4%) developed a mild primary infection..

- >  2 patients (5.4%) developed a mild primary infection.

Author Response

Dear Reviewer,

thank you once again for your further detailed review.

We hope we have now addressed all of your concerns properly. The word usage should be consistent, the font size for each author and affiliation was corrected.

We tried to use the same abbreviations throughout the whole text. The numbers of Figure 1 were corrected. 

Warmest regards,

Enke Grabhorn
